# Proposed Conceptual Framework to Design Artificial Reefs Based on Particular Ecosystem Ecology Traits

**DOI:** 10.3390/biology11050680

**Published:** 2022-04-28

**Authors:** Luis Carral, María Isabel Lamas, Juan José Cartelle Barros, Iván López, Rodrigo Carballo

**Affiliations:** 1Escuela Politécnica Superior, Universidade da Coruña, C/Mendizábal, 15403 Ferrol, Spain; isabel.lamas.galdo@udc.es (M.I.L.); juan.cartelle1@udc.es (J.J.C.B.); 2Área de Ingeniería Hidráulica, Universidade de Santiago de Compostela, C/Benigno Ledo St. 2, 27002 Lugo, Spain; ivan.lopez@usc.es (I.L.); rodrigo.carballo@usc.es (R.C.)

**Keywords:** artificial reefs, ecosystem ecology perspective, autotrophic resource pathways, habitat management, stability analysis, computational fluid dynamics

## Abstract

**Simple Summary:**

Artificial reefs improve the yield of artisanal fishing grounds by creating habitats with enhanced productivity, similar to those of natural reefs. For an adequate definition of an artificial reef, all its pathways of interaction with the ecosystem must be quantitatively assessed. In this way, the final design will lead to an increase in the production of services. Therefore, this study presents the MEEM tool as well as the AREIT index. Through the former, it is possible to obtain different reef solutions, while the index allows the user to select the best alternative by taking into account the specific situation of the ecosystem to be enhanced. The AREIT index facilitates the decision-making process by means of a three-dimensional graphical representation, based on the three aspects of greatest relevance for the ecosystem: energy, nutrient circulation and the presence of cavities for the shelter of species, in particular those of fishing interest. The definition process ends with the application of a procedure for calculating the stability of the final design under the action of waves and currents.

**Abstract:**

Overfishing and pollution have led to marine habitat degradation, and as a result, marine fisheries are now in decline. Consequently, there is a real need to enhance marine ecosystems while halting the decline of fish stocks and boosting artisanal fishing. Under these circumstances, artificial reefs (ARs) have emerged as a promising option. Nevertheless, their performance is traditionally assessed years after installation, through experimental and field observations. It is now necessary to adopt an alternative approach, adapting the design of artificial reefs to the specific characteristics of the ecosystem to be enhanced. In this way, it will be possible to determine the potential positive impacts of ARs before their installation. This paper presents a general and integrated conceptual framework to assist in the design of AR units by adopting an ecosystem ecology (EE) perspective. It consists of three main parts. In the first one, starting from an initial geometry, EE principles are used to include modifications with the aim of improving autotrophic resource pathways (additional substrata and increased nutrient circulation) and leading to a habitat enhancement (more shelter for individuals). The second part of the framework is a new dimensionless index that allows the user to select the best AR unit design from different alternatives. The potential impacts on the ecosystem in terms of energy, nutrient cycling and shelter are considered for such a purpose. Finally, a general hydrodynamic methodology to study the stability of the selected AR unit design, considering the effect of high waves under severe storms, is proposed. The framework is applied through a case study for Galician estuaries.

## 1. Introduction

Research on artificial reefs (ARs) is a multidisciplinary activity that has attracted the attention of a great number of researchers all over the world [1,2]. In fact, the Web of Science database contains more than 200 scientific papers on ARs published in more than 100 different journals and belonging to more than 50 different research categories, in 2020. Throughout this paper, ARs are understood as man-made structures specifically designed and manufactured to fulfil at least one of the functions performed by natural reefs. ARs are usually modular structures with holes, with concrete being the most common material employed in their construction, and the one considered in this study. In particular, ‘green’ concretes have been proposed in previous works [3,4].

Overfishing and pollution have led to a marine habitat degradation, endangering the exploitation of its resources [5]. On the other hand, if current population growth is maintained, demand for fishery resources is expected to rise due to, among other things, the corresponding increase in per capita consumption of fish. Nevertheless, marine fisheries are in decline, which is further evidence of the unprecedented global environmental crisis we are facing [6]. In this context, the need to enhance marine ecosystems while halting the decline in fish stocks and boosting artisanal fishing is evident.

As pointed out by Carral et al. [7] new measures are required to enhance coastal ecosystems. In this regard, the installation of ARs has emerged as a promising option for both providing habitat for marine species and promoting sustainable fisheries [8]. Furthermore, ARs can also provide alternative ecosystem services, since they also attract recreational divers [9].

On the other hand, according to Daza-Cordero et al. [10], hard substrata are notably superior to soft ones in terms of productivity, diversity and the abundance of living organisms. It is a proven fact that some marine structures, such as jetties [11] and oil platforms [12], function as natural reefs generating high densities of aggregating fish in shallow coastal areas and deep waters, respectively. However, some authors have pointed out the importance of designing artificial reefs specifically for the ecosystems intended to be enhanced [8,12].

The performance of artificial reefs was traditionally assessed through population and community ecology through approaches based on experimental and field observations to quantify fish density, biomass or composition, among other indicators linked to the species of interest. Nevertheless, some authors (Jayanthi et al. [13], Layman and Allgeier [8] and Liversage [14]) highlighted the need to design ARs adopting an ecosystem ecology (EE) perspective which enables identification of the factors that limit the net primary production (NPP) of a particular ecosystem. Once this has been done, the design of the ARs can be aimed at increasing NPP and nutrient cycling. In this way, it will be possible to determine the extent to which the ARs can contribute to enhancing a specific ecosystem before their installation. According to Lindeman [15], the secondary productivity is, in most ecosystems, linked to NPP and how energy and biomass pass through the different trophic levels. Consequently, by adopting an EE perspective in the design of ARs, it will also be possible to estimate the impact they can have on secondary productivity.

In this line, Geider et al. [16] state that NPP at all trophic levels depends mainly on three factors: (i) substratum availability, (ii) nutrient availability and (iii) light penetration, although the relative importance of each of these factors varies from one ecosystem to another. Therefore, the design and installation of ARs must ensure new autotrophic resource pathways by (i) increasing nutrient availability or (ii) providing additional substratum (surface area) for sessile organisms, algae and plants [8]. These two strategies can occur together or independently. On the other hand, ARs are beneficial for their positive impact on nutrient availability as well as for providing shelter [17]. In fact, in addition to the increase in surface area previously indicated, ARs should present vertical planes as well as holes and cavities. This will also serve to improve nutrient circulation [18]. The number of studies directly addressing the relationship between the design of ARs and their potential impacts on the autotrophic communities is limited [8]. In fact, most of existing studies focused on autotrophs examine the community composition [19,20,21], with much less attention paid to changes in NPP or in other processes of the ecosystems. In this regard, Lemoine et al. [22] suggest that there is a wide range of actions that can be taken in the design of ARs in addition to the common ex-post studies previously mentioned. They promote the design of specific ARs by taking into account all the ecosystem processes to be improved. In this connection, Sedano et al. [23] analysed the ecological effects of artificial substrata on marine environments and their biota. They highlighted the necessity of studying other factors (biological interactions and hydrodynamics or water currents, among others) at the time of designing an artificial reef with the aim of globally understanding the particularities of the ecosystem to be enhanced.

Following the recommendation of Lemoine et al. [22], Layman and Allgeier [8] adopted an EE perspective [24] and studied how the ecological mechanisms of ARs can serve to maintain secondary production through NPP, decomposition and trophic interactions. Also with the objective of analysing the potential benefits of ARs on fauna, biomass and diversity, Lima et al. [25] developed two artificial reef multimetric indices (ARMIs). Both indices consider trophic structure, vulnerability, economic importance and structure of fish assemblages. One of the indices (ARMIr) is for use on a global scale, while the other one (ARMIe) measures the impact of the AR in comparison with a control area in the same region. The authors considered a case study in Rio de Janeiro, Brazil.

Carral et al. [26] developed a methodology in order to determine the best area for installing ARs in a specific estuary, by taking into account parameters such as the geomorphological, hydrodynamic and bathymetric characteristic of the area; the distance to the closest harbour as well as different production and economic factors. The method proposed by the authors is based on a geographic information system (GIS) and it was applied to an estuary in the northwest of the Iberian Peninsula. Carral et al. [27], based on the results obtained with the previous method, combined two 2D hydrodynamic models for the same estuary with the objectives of determining the appropriate distance among all the AR units that constitute an artificial reef group, and at the same time, improving the circulation of the nutrients along the reef cavities.

Despite the most recent efforts adopting novel perspectives in the design of ARs, there is still considerable scope for improvement, in particular in terms of EE.

Taking into account the gaps in current knowledge, the objectives of this work were: (i) to propose a conceptual framework to assist in the design of artificial reefs according to the specific characteristics of the ecosystem to be enhanced, (ii) to propose a comparative index (AR-Ecosystem Index Transformation or AREIT) and (iii) to develop a hydrodynamic methodology to evaluate the stability of AR units considering high waves and surfaces.

The conceptual framework will be based on EE principles; therefore, the relationship between the design of an artificial reef and the secondary production of the ecosystem [28] through the increase in NPP in all autotrophic pathways will be taken into account. On the other hand, the AREIT index will assist decision makers in selecting among different AR alternatives. It measures the relative impact on the ecosystem in terms of energy, nutrient cycling and shelter (habitat modification). Consequently, it consists of three partial indices (Section 2.3) which, if necessary, could also be used separately. As part of this study, the AREIT index will be applied to a simple case study for Galicia (in the northwest of Spain).

## 2. Materials and Methods

### 2.1. Conceptual Framework for the Design of AR Units

The framework proposed in this study consists of three main parts as can be seen in Figure 1. The first arises around the Marine Ecosystem Ecology Model (MEEM). It starts from an initial basic geometry (a cube, a prism or a pyramid, among other possibilities) with specific dimensions and weight. The starting geometry takes into account all logistical issues (manufacturing and transport, among others) as well as all the necessary information (current velocity, seabed characteristics, and the effects of the wind, tide, waves and water discharges from rivers) at the location of the AR units.

Together with the input information (Figure 1), MEEM uses ecosystem ecology (EE) principles (Section 2.2) to modify the initial geometry, mainly through the inclusion of holes and nest cavities. A nest cavity is a blind hole or a surface indentation and, unlike other type of holes, it is not intended to connect different AR unit spaces but to provide more shelter [27,29]. The number of holes and nest cavities as well as their dimensions are conditioned by the structural stresses that the AR unit will suffer during its life cycle, from the manufacture to the installation process. Nevertheless, as the result of following EE principles and taking into account the structural constraints, a wide range of different AR unit designs may arise. For such a purpose, the AREIT index is used as the second main part of the proposed framework.

As will be explained later in Section 2.3, this index preferably serves to measure the performance of those designs that contribute to the improvement of the ecosystem in terms of energy, nutrient availability and habitat. If a specific AR unit design does not contribute to one of the three previous dimensions, the complete AREIT index may not be used. The generation of new designs finishes when the result provided by this metric for one of the alternatives is good enough for the intended purpose. This usually means obtaining a sufficiently high overall index or obtaining a certain value for one or more of the three partial indices that make up AREIT (Section 2.3). Obviously, the iterative process of generating and assessing new AR unit designs can be transformed into a single or multi-objective mathematical constrained optimisation problem. Nevertheless, this is out of the scope of this study.

When the desired AR unit design is obtained, the third part of the framework comes into play, that is, the stability study. This analysis may lead to the inclusion of a concrete slab with a certain height (H in Figure 1).

It is important to remark an alternative use of the proposed framework that affects the nomenclature of Figure 1, and that has already been indirectly introduced. There can be real situations in which the decision maker only needs to consider one or two of the three dimensions included in AREIT. This can be the case, for example, when an ecosystem presents problems in terms of energy or nutrient circulation, while additional shelter is not needed. In such cases, AREIT partial indices must be used as a benchmark for comparison among designs, instead of the overall index. It would also be possible to create a modified AREIT index by taking into account only the dimensions under study. In both circumstances, the selected index, as in the general case described here, will only serve to compare designs that contribute to the dimensions under consideration. The reader can find in Section 2.3 more information about AREIT and its corresponding partial indices.

The remainder of Section 2 is organised as follows. Section 2.2 provides information about the EE principles as well as an overview of the MEEM model. The AREIT index is described in Section 2.3 and, finally, the stability methodology is presented in Section 2.4.

### 2.2. Ecosystem Ecology (EE) Principles and the Marine Ecosystem Ecology Model (MEEM)

Food web dynamics are an important part of ecosystems. An artificial reef can create, on a nearly deserted area, a system with its entire food web, functioning as an “energy trap”. Consequently, nutrients are recycled and incorporated into the food web, instead of being deposited in the ocean floor for thousands of years [10].

In other words, a submerged structure interacts with the aquatic environment, creating new attachment locations. Therefore, sessile organisms in a planktonic larval stage circulating through sea currents find new attachment surfaces. In this way, artificial reefs are first colonized by algae and sessile invertebrates, and their energy is consumed by vagile organisms such as crustaceans and herbivorous rockfish. These and other organisms will spawn at the artificial reefs and, predators and top predators will also find a place to shelter, feed and reproduce. As a result, artificial reefs increase both the energy and nutrient availability, boosting the food web of a specific ecosystem (Figure 2).

Plants exercise control over the energy of the ecosystem, since they delimit the amount that is received. Consequently, the productivity of marine ecosystems may be limited even if there is high nutrient availability, particularly in cases where there is not enough substrate for benthic organisms to settle [8]. Under these circumstances, artificial reefs may be the solution to increase secondary production as indicated by Layman and Allgeier [8] and by Salamone et al. [30]. Similarly, substrate composition can have a major impact on the development of marine biofilms and, therefore, on the diversity of the final fauna [31,32,33,34]. In fact, Salamone et al. [30], after comparing the fungal diversity on four different ARs in North Carolina (USA), pointed out that the type of substrate could be a determining factor for the floral an faunal community composition of the ecosystem.

On the other hand, artificial reefs cause changes in the velocity field of the water [35], favouring nutrient circulation [35,36]. More specifically, the inflow of water impinging on the AR separates, so that upwelling and back eddy effects take place, giving rise to a vertical exchange of water.

One of the reasons fish aggregate around ARs is local upwelling and back eddy effects [37,38,39]. Many of them remain close to the AR during the day, feeding in adjacent seagrass beds [11,40]. Their excretion and egestion serve to increase the nutrient availability (mainly nitrogen and phosphorus) [8]. In this way, fish also enhance NPP [41].

In describing the foundations of the MEEM model, it is necessary to consider a baseline design for the AR unit. In this case, taking into account the logistical processes of the management and enhancement of an estuary in Galicia by installing AR units [42], a cube with edges of 1.5 m appears to be a good option. The cube (upper prism) may need a concrete slab for stability purposes (third part of the conceptual framework, Section 2.4). This slab has no biological objectives and, consequently, it can be excluded from the AREIT index.

As previously indicated, MEEM is aimed at guiding the design of an AR by improving the two autotrophic resource pathways that an ecosystem has (additional substrata (algae and plants) and increased nutrient circulation (phytoplankton)), and also by leading to habitat enhancement.

According to Bohnsack [43] artificial reefs can give rise to an increase in fish biomass (secondary resource) through different general pathways (Table 1). At the same time, several authors point out that fish aggregation around artificial reefs depends on physical parameters such as the area, type of material, roughness, verticality, and nest cavities (number, size and shape) [44,45,46] (Table 1). Consequently, all these parameters must be taken into account at the time of designing an AR unit from an EE perspective. On the other hand, it is also necessary to consider the strategies identified by Layman and Allgeier [8] for increasing the secondary production through the development of new autotrophic resource pathways (Table 1).

From Table 1, it can be concluded that, in addition to the aforementioned functions of ARs in terms of energy conversion and nutrient transfer, it is necessary to provide the AR unit with nest cavities. They serve as shelter and as a place for spawning and feeding for several species of fishery interest. This is a habitat modification. Table 2 shows the main determinants of an AR unit as well as their significance in terms of energy, nutrients and new habitat for the ecosystem and summarises the foundations of the MEEM model that the designer or decision maker must take into account at the time of using the proposed framework.

Vertical surfaces increase the available substrata, while at the same time collect less sunlight than horizontal planes. Including an interior cavity (upper central hole) with an opening at the top can serve to increase the sunlight collection. Furthermore, lateral holes (connecting the interior of the AR unit with the exterior) in vertical surfaces reduce the substrate availability but favour nutrient circulation inside the AR. Lemoine et al. [22] state that concrete AR modules are the most appropriate alternatives for habitat restoration, since they are similar to rocky reefs resulting in comparable biomass and fish abundance and composition. Furthermore, concrete AR modules are chemically and physically resistant to marine conditions, an indispensable characteristic according to Lukens and Selberg [47]. In a similar line, the pH of the base material should be similar to that of the local habitat, that is, between 7.4 and 7.6 for seawater [48,49]. Large variations in pH (both acidic and basic) can cause negative impacts on marine organisms [50] and cement can only cause minor pH variations, as pointed out by Matsunaga et al. [51].

Table 2 provides general information that must be considered during the design of an AR unit following the EE principles. It is not possible to provide a more specific methodology with concrete steps to be followed, since the number and the dimensions of the holes and nest cavities, among other parameters, are site dependent.

### 2.3. AR-Ecosystem Index Transformation (AREIT)

The index proposed in this section measures the potential transformation that the ecosystem may experience after the installation of one AR unit in comparison with a basic or reference design. It consists of three partial indices: *EM* (energy modification index), *NM* (nutrient modification index), and *HM* (habitat modification index), Equation (1).
(1)AREIT=EM+NM+HM

In Equation (1), it has been assumed that the three dimensions or partial indices present the same weight or relative importance. Nevertheless, this may vary from one ecosystem to another, depending on the particular needs. Consequently, Equation (1) can be easily modified to consider this possibility, by including the weight of each partial index. On the other hand, it is important to remember that the application of this index is limited to those cases in which the reference design contributes to the improvement of the ecosystem in terms of energy, nutrient availability and habitat. The *AREIT* index is a comparative, dimensionless parameter and a higher value is associated with a higher positive impact on the ecosystem in comparison with a reference design. The same is also applicable to the three partial indices. More information on how to interpret these parameters will be provided later in this section.

The impact on the energy of the ecosystem depends on the primary producers and, consequently, on the AR attachment surfaces exposed to sunlight. Therefore, *EM* is estimated through Equation (2):(2)$$EM=\frac{S_{v}\cdot f_{ev}+S_{h}\cdot f_{eh}}{S_{vr}\cdot f_{ev}+S_{hr}\cdot f_{eh}}$$
where *S_v_* is the total area of the external and internal vertical surfaces of the AR unit exposed to sunlight, measured in cm^2^. Similarly, *S_h_* is the total area of the external and internal horizontal surfaces of a specific AR design, expressed in the same units. *S_vr_* and *S_hr_* are analogous to *S_v_* and *S_h_* for the basic or reference AR design. Moreover, *f_ev_* and *f_eh_* are the dimensionless exposure factors for vertical and horizontal surfaces and they adopt the values 0.5 and 1, respectively. It is important to note that *EM* takes a value of 1 for the reference AR design. Accordingly, *EM* values below 1 are not desirable. On the contrary, the higher this partial index is compared to the reference case (*EM* = 1), the better the AR design is in terms of its contribution to the ecosystem energy increase. The same is also true for *NM* and *HM* indices in terms of nutrient and habitat enhancement. On the other hand, Equation (2) was designed for ARs that do not present slope surfaces. Nevertheless, it can be easily adapted to designs with inclined planes by including new exposure factors varying between 0.5 and 1, depending on the inclination angle.

The positive impact on the nutrient transfer among the organisms of the ecosystem depends on the modification that the AR unit introduces in terms of nutrient cycling. In other words, both the upwelling and back eddy effects come into play at the time of defining the *NM* index. The reader should bear in mind these both effects are correlated, since upwelling always leads to back eddy formation. Therefore, by taking into account the upwelling effect, the back eddy phenomenon is also implicitly considered. Equation (3) is used for estimating *NM*:(3)$$NM=\frac{S_{upwelling}}{S_{upwelling\_r}}$$

*S_upwelling_* is the total area in cm^2^ of the AR surfaces that contribute to the upwelling and back eddy effects, that is, vertical ones, from which the area of the lateral holes is subtracted. Lateral holes communicate the interior of the AR unit with the exterior, generating a permeability effect that reduces upwelling and back eddying. Consequently, Equation (4) is used for calculating *S_upwelling_*:(4)$$S_{upwelling}=S_{v}\cdot f_{rv}-\overset{n}{\underset{i=1}{\sum }}S_{lh,i}\cdot f_{r,i}$$

In previous equation, *f_rv_* is the roughness factor for the vertical surfaces. This factor, as its name suggests, measures the roughness of the material and depends on its type and finishing. The same roughness factor is assumed for all the vertical surfaces, since all of them are usually made of the same material with the same finish. Nevertheless, Equation (4) can be easily modified for other different cases. Furthermore, *n* is the total number of lateral holes. There can be holes with different dimensions. Consequently, *S_lh,i_* is the surface in cm^2^ of each one of the holes “*i*” with *f_r,i_* its corresponding roughness factor. As was the case with Equation (2), Equation (4) is valid for AR designs in which all surfaces are vertical or horizontal. For those cases with sloping surfaces, this equation must be modified, by taking into account the sloping surface areas with their corresponding roughness factors. In this sense, the closer a specific surface approaches verticality, the higher its contribution to upwelling and back eddy effects. Therefore, it will be necessary to consider and additional dimensionless factor to reflect this fact, varying between 0 (horizontal surface) and 1 (vertical surface). The computation of *S_upwelling_r_* is analogous but, this time, for the reference design.

Finally, the habitat modification index (*HM*) is estimated by applying the following equation:(5)$$HM=\frac{S_{nest\_cavities}}{S_{nest\_cavities\_r}}$$

As can be deduced, it depends on the number of nest cavities and their corresponding areas. There can be nest cavities with different dimensions. As a result, *S_nest_cavities_* (cm^2^) is calculated from Equation (6), where *nc* is the number of nest cavities and *S_n,j_* is the surface area of nest cavity *j*. *S_nest_cavities_r_* is estimated by applying Equation (6) but, this time, for the reference AR design.
(6)$$S_{nest\_cavities}=\overset{nc}{\underset{j=1}{\sum }}S_{n,j}$$

In light of the above, an *AREIT* index under 3 (equal weightage) implies that the AR design presents an overall worse performance than the reference one. By contrast, values higher than 3 mean that the AR design generates a higher positive impact on the ecosystem (EE perspective) than the reference case. It is important to note the *AREIT* index is a comparative parameter. In other words, it measures how good a specific design is compared to a reference one with the same volume. Comparing two designs with very different sizes and volumes using this indicator can lead to wrong conclusions. Furthermore, the ultimate objective of this index is to help the decision maker identify the design with the greatest positive impact on the ecosystem per unit of volume (occupying the same space).

Furthermore, the reader should bear in mind the advantages of estimating the three partial indices separately. There can be real marine ecosystems with improvement needs in only one or two of the three requirements (energy, nutrients and habitat) considered in *AREIT*. Therefore, by estimating the partial indices on an individual basis, it will be possible to identify the design that most improves the specific weakness of the ecosystem. As a result, the graphical representation in a three-dimensional space with three axes (each one linked to a specific partial index) is proposed (Figure 3).

The *EM* and *NM* axes allows the user to quickly identify the improvements achieved according to the two autotrophic resource pathways that delimit the primary productivity. The remaining axis provides information on the new habitat possibilities for sheltering, feeding and reproducing. The position of the plane defined by the three partial indices of a specific AR design in comparison with the plane of the base case serves the same purpose.

The reader should bear in mind that there are other ecological factors and processes as well as other social, economic and environmental issues that can affect the design of AR units and that are not considered in *AREIT*. However, as previously stated, the conceptual framework presented in this study is a starting point for the design of AR units according to EE principles. On a practical level, a compromise solution between the design provided by the framework and other potential factors will need to be found. Despite the advantage of being able to predict the potential impact of an AR unit design on the ecosystem through *AREIT*, experimental and field observations are still valuable, and they can serve to validate and, if necessary, to modify the index proposed.

### 2.4. General Methodology for Studying AR Stability

The study of the stability of ARs should be based on a thorough analysis of their interaction with the hydrodynamics at the site where they are planned to be installed, with special consideration of extreme conditions with large wave heights, which are usually regarded as the major forcing agent in the design of coastal structures, and in particular of aquaculture farms [52]. This is also the case when designing ARs given that despite them being usually located in sheltered areas with large biological production, such as estuaries, they may be subject to high waves in extreme events. In order to illustrate this, in Figure 4, the resulting distribution of the significant wave height, *H_s_*, in the Ría de Ares-Betanzos (Galician estuary) under deep-water wave conditions for a return period *T_r_* = 20 years and NW direction is plotted. It can be observed that under these conditions, extensive areas of the outer and middle estuary present *H_s_* values greater than 5–6 m, which in turn would correspond to maximum wave heights, *H_max_*, of about 10 m.

As a result, the analysis of the stability of an AR configuration considering wave action should be divided in two phases of the same process: (a) a hydrodynamic characterization where the hydrodynamics generated by the most relevant forcing agents under extreme conditions are assessed, amongst which the estimation of the so-called design wave (an extreme sea state or individual wave defined depending on the type of coastal structure or AR considered) at the AR site is usually of paramount importance; and (b) a force analysis in which the behaviour of the AR under the action of the design conditions defined in (a) is addressed. In the following paragraphs, key aspects and a possible procedure for developing an accurate AR stability analysis are described. For the sake of clarity, in Figure 5 a flowchart of the different available approaches is provided to guide the reader through the description presented.

As previously stated, waves are usually the most relevant hydrodynamic agent under extreme conditions in most areas of interest for AR operation. Reliable estimation of the wave field, and on its basis, of the design sea state or wave for a specific coastal structure, requires analysis of a large dataset of sea states. Field data can be used to develop a thorough study at specific locations where a buoy has been in operation over extensive periods of time. Unfortunately, instrumental data is not usually available at the locations of interest for AR installation. In this context, once the most suited locations for AR operation are identified, a buoy or ADCP (acoustic doppler current profiler) could be installed at these sites; however, the resulting dataset would cover a limited period of time, which would not be adequate to determine the design wave conditions. In order to overcome this, meteorological agencies run global wind–wave models, which consider wind field estimations for numerically generating long-term wave data over large areas, leading to an accurate description of the wave field at sites where instrumental data is not available. Unfortunately, these models usually do not consider the interaction of the wave field with the bathymetry [53], and underestimate extreme sea states; thereby they do not provide reliable estimations of the wave field in nearshore areas for determining the highest waves. Considering the limitations of the current datasets available in nearshore coastal areas, the procedure to be developed for estimating the design wave conditions for an AR should include the implementation of a downscaling procedure, usually based on the application of high-resolution local numerical models to propagate offshore wave data—either provided by large-scale models or registered by deep-water wave buoys—towards the selected coastal locations.

In this regard, as previously stated, a large proportion of ARs are intended to be located in coastal areas with large biological production, such as estuaries, generally sheltered to a certain extent from wave action; however, during severe storms, as stated, wave action is usually the most relevant hydrodynamic forcing agent in these areas. This results in the need to consider two additional major aspects when defining the downscaling procedure to be implemented.

First, resulting from the usual coastal configuration of estuaries and the depth-limited areas of interest for AR installation, the highest nearshore waves at the AR site may not correspond to the highest offshore waves, thereby requiring the analysis and selection of many conditions to be propagated towards the area of interest, for which the implementation of wave spectral numerical models is the common valid procedure. Spectral models compute the wave spectrum evolution by solving the action balance equation, Equation (7):(7)$$\frac{\partial }{\partial t}N+\nabla \cdot \left(\vec{C}N\right)+\frac{\partial }{\partial \theta }\left(C_{\theta }N\right)+\frac{\partial }{\partial \sigma }\left(C_{\sigma }N\right)=\frac{S}{\sigma }\;$$
where *N* is the density of wave action, *t* stands for the time, *C* represents the propagation velocity in the geographical space, *C_θ_* and *C_σ_* the propagation velocities in the wave direction, *θ*, and relative frequency, *σ*, and *S* stands for the source or sink term.

Selection of the wave conditions to be analysed should be conducted using methodologies developed ad hoc, such as those provided by IHCantabria [53,54], or adaptations of others based on the energy bin concept [55]. The resulting data could be also incorporated into a geographic information system (GIS) to be used at an earlier stage to identify the most suitable areas for installing ARs. In addition, the bathymetric configuration of these areas, including sudden depth changes and obstacles, may well lead to abrupt wave propagation processes, such as diffraction or wave breaking. In this case, the implementation of high-resolution spectral models (with a spatial resolution in the order of *L*×10−1, where *L* is the wavelength) is required [56]. Another aspect to be considered relies on the fact that spectral models may not provide accurate results in the case of abrupt changes in water depth in very shallow areas, as could be the case at specific sites for AR operation. In this case, spectral models could be used to determine the design conditions in the surroundings of the AR location, and these conditions used as input for computational fluid dynamics (CFD) models (part (b)) to obtain more reliable wave characteristics at the site of interest.

Finally, it is important to note that the forces exerted over the AR configuration by nearshore currents either generated by the tide, river discharges, wind or baroclinic flows are expected in most cases to be virtually negligible compared to those caused by the action of an extreme wave. However, certain areas of interest for AR operation could be subject to strong currents—usually tidal currents—which should also be considered in the stability analysis, both for the definition of the design hydrodynamic conditions to be analysed in part (b), and for the analysis of the complex wave–current interaction process, which could also affect the definition of the design conditions. To this end, the implementation of shallow water numerical models (coupled with spectral models) has been shown to provide accurate results of the flow field in coastal regions, and more specifically in estuaries.

Once the design conditions are determined at the site selected for the installation of an AR, its stability should be analysed (part (b)), and on this basis, actions defined (e.g., the deployment of a flagstone or concrete slab) if necessary, to ensure that the reef does not overturn or slide. To this end, several approaches of different complexity ―and therefore of computational effort― are available, whose selection depends on the level of accuracy required. In this document we identify three procedures (Figure 6): (i), (ii) and (iii) (from higher to lower complexity and accuracy).

An accurate analysis of the stability (i) requires the implementation of CFD modelling, in which the selected AR configuration is considered, subject to the action of the design conditions previously defined. In addition, in the case of possible significant modifications in the wave field in the surroundings of the AR resulting from the presence of a complex bathymetric configuration and reduced depths, which may not be accurately captured by the spectral propagation model, the bottom configuration in the proximity of the AR can be easily represented in CFD models in the form of a bottom slope. This type of model demands a high computational cost resulting from the different orders of grid size that are necessary to implement (a reduced grid size for representing the holes in the AR in comparison with the grid size required for the rest of the computational domain). Furthermore, analysis of the behaviour of an AR ideally requires the generation of many waves (200 or more) in order to decrease the sampling variation of the wave statistics [57] and to increase the probability of occurrence of the highest wave heights of the sea state, which in turn would lead to an even higher computational cost. On these grounds, with the aim of reducing the computational effort while maintaining the accuracy of the results, and considering the characteristics of most of AR configurations, the analysis of the stability of the AR under the action of the highest wave during the design sea state could be a valid approach. A key point in this aspect is the generation of the highest wave by using the appropriate wave theory. To this end, there exist different wave theories [58] whose adequacy depends on the type of wave to be analysed. Although there is not a general rule to be followed, the most common highest waves to which ARs are likely to be subjected correspond to highly nonlinear waves in shallow water areas occurring during severe storms near the coast, waves which are usually well approximated through the Cnoidal wave theory [59].

The previously described methodology has emerged as the most accurate approach; however, it consists of a complex procedure which can be significantly simplified when conducting preliminary studies. First, in procedure (ii) a CFD model can be used to accurately compute the characteristics of the design wave conditions at the location of interest instead for the analysis of the hydrodynamics–AR interaction. In this case, once the hydrodynamic characteristics are computed at the AR location, the stability analysis can be carried out through empirical or analytical formulations traditionally used in coastal engineering for submerged structures, or designed ad-hoc for an AR configuration [60]. Among the different formulations, the computation of the resulting forces by considering the Morison formulation is a common procedure in coastal and aquaculture engineering [61]. An even simpler approach (iii) does not resort to CFD modelling at any of the different steps (either wave propagation, hydrodynamic characterization or hydrodynamics–structure interaction). This approach consists of the straightforward analysis of the AR stability by considering the empirical or analytical formulation used in (ii) where the required hydrodynamic data is obtained from the analysis in (a) and, if required, from the direct application of a specific wave theory formulation (e.g., linear wave theory for determining the orbital wave velocities).

It is important to highlight that in the case of areas subject to strong currents, as previously explained, they should be incorporated in the definition of the design conditions and considered in the force analysis following the aforementioned procedures.

## 3. Results and Discussion

This section is divided into two subsections. In Section 3.1 an AR unit design (upper prism) is proposed as an example by following the foundations of the MEEM model (Table 1 and Table 2) for Galician estuaries. In Section 3.2, the design proposed is compared to a reference or base one through the *AREIT* index.

### 3.1. Example of an AR Design Proposal following the Foundations of the MEEM Model for Galician Estuaries

Figure 6 shows the design (upper prism) proposed after following the EE principles (MEEM model). Its weight is about 5 tonnes, and the design is based on a cube block (excluding the possible concrete slab with height H) that appears as a promising initial geometry for enhancing Galician estuaries [26,42].

The cubic shape was used since this provides a proper upwelling, as will be shown below. This cube has several cylindrical holes. The cylindrical shape of the holes was adopted for constructive reasons, since the concrete manufacturing of a cylindrical hole is easier than other shapes [62]. A 600 mm diameter interior hole connects the exterior with the interior in order to provide light as well as a communication for nutrient reception. Two lateral holes (250 and 450 mm in diameter) also communicate the exterior with the interior. This configuration guarantees circulation into the interior part of the AR. The nest cavities are cylindrical holes placed around the faces. These nest cavities have different sizes according to the species that will colonize the AR. The small vertical holes are included for the installation into the sea. During the installation, the AR is suspended through ropes which are introduced in these suspension system holes. Although it is out of the scope of this study, the proposed AR unit design is capable of withstanding all the mechanical stresses to which it will be subjected during its life cycle. In fact, a comprehensive structural calculation was carried out elsewhere [62]. A specialised company was responsible for developing a detailed constructive strategy as well as for designing transport and draught operations. Including more holes and cavities or increasing the size of existing ones will increase the likelihood of breakage of the AR units. Furthermore, the dimensions of the nest cavities are adapted to Galician marine fauna. On the other hand, the MEEM model could provide very different designs from the one included in this section, depending on the site-specific characteristics of the ecosystem to be enhanced, including a different starting design.

Coming back to the design of Figure 6, the outer sides of the prism favour energy conversion (additional substrata) since plants can absorb the incident sunlight. At the same time, vertical sides enhance the circulation of nutrients. Lateral faces provide surfaces for the attachment of benthic algae, constituting the base of the food web.

A CFD model was developed, validated with experimental measurements in a previous work [27]. This is based on the conservation equations of mass and momentum. Turbulence was treated through the k-ε turbulence model. The free software OpenFOAM [63] was employed. The current velocity was obtained from a circulation model developed elsewhere [27], and it was established as 0.08 m/s. A 25 × 10 × 5 m computational domain was employed. In order to save computational time, symmetry was applied and only half of the geometry was treated. The computational mesh is shown in Figure 7, as well as the boundary conditions. As can be seen, tetrahedral elements were employed, and the mesh size was refined around the reef. Figure 8 includes the velocity field (m/s) obtained by the CFD model.

As can be seen in Figure 8, an upwelling is created above the AR. This upwelling displaces nutrients from the lower part to the upper part of the AR, which is very positive to promote nutrient circulation and to provide food for the organisms located at the upper parts. At the same time, an eddy is produced at the back part of the AR. This back part is also critical since it is not directly exposed to the current velocity and thus is susceptible to becoming a nutrient-poor region. Nevertheless, the back eddy promotes nutrient circulation which improves the food availability and thus makes this an appropriate zone to allocate nest cavities. Both upwelling and back eddy effects provide a vertical exchange of water. According to this, the proposed design will attract fishes due to high nutrient availability.

Regarding the communication along the AR, the velocity field illustrated in Figure 8 also shows that the lateral holes communicate the central hole with the outside of the artificial reef. This communication between the central and lateral holes allows nutrient circulation in the interior of the AR unit. Besides, the central hole provides an additional surface area for the attachment of organisms, where the velocity field is appropriate. This incorporates extra vertical planes as well as confined or semi-enclosed spaces and transit ones which increase the spatial diversity of the substrate.

The proposed design favours the exposure to sunlight as well as current and sedimentation processes. These factors strongly influence both the settlement of individuals and the evolution of the community [61]. Rouse et al. [28] demonstrated that productivity rates of certain organisms are considerably higher on artificial reefs constructed from complex blocks, such as the proposed design, than on reefs made from simple blocks.

New habitat is provided by the inclusion of nest cavities (20 or 30 cm diameter) in the lateral faces of the cube. They serve as nests for cephalopods and crustaceans. Furthermore, these cavities imply that a greater proportion of the AR unit serves as habitat for benthic species, which can result in higher productivity [28]. In fact, nest cavities can be used by different fish species and marine organisms such as lobsters or sea urchins, enhancing colonisation through the settlement of sessile benthic organisms.

### 3.2. AREIT Results: Comparison between the Design Proposal for Galician Estuaries and a Reference One

The final AR design presented in Section 3.1 was assessed through the *AREIT* index. As previously explained, a reference design must be considered for such purpose. The starting design presented in Section 2.2 is not valid since it does not contribute to the *HM* partial index (although it contributes to both *EN* and *NM* indices). Therefore, another alternative design was considered as reference for this section. It is also a cube with an edge of 1.5 m but, in this case, with a reduced number of nest cavities (*HM* contribution), as can be seen in Figure 9 (right AR unit). Both designs, where necessary, must utilise the same concrete slab for stabilisation purposes. Consequently, the differences in their contribution to the ecosystem enhancement lie in the upper prism. Table 3 contains the most relevant information of both designs as well as the values that the main parameters take.

From the information included in Table 3, it is possible to estimate the *AREIT* index as well as the corresponding partial indices for the final design. As explained in Section 2.3, the reference alternative obtained a value of 1 for all the partial indices (*EM* = 1, *NM* = 1 and *HM* = 1). Consequently, its AREIT index is equal to 3. The results for the final design in terms of *EM*, *NM* and *HM* should be higher than 1 in all cases. This would mean that it is better than the reference in the three dimensions considered in this study. In this case, this is true as can be seen in Figure 9.

In fact, the upper central hole together with the lateral ones generated a *NM* value of 1.2, which is 20% higher than the reference index. In other words, the proposed design considerably improves nutrient circulation in comparison with the base case. Furthermore, this is achieved by maintaining (even slightly increasing) its capacity as a support for vegetal substrate (*EM* = 1.02). At the same time, the positive impact in terms of habitat modification that the proposed design produces is considerably higher than that of the reference case (*HM* = 1.67). The *AREIT* index for the proposed AR unit takes a value of 3.89. This figure, in comparison with the reference value of 3, gives the decision maker a quantitative idea of the extent to which the proposed design better enhances the ecosystem. Consequently, it can be concluded that the proposed index is suitable for comparative purposes with the considerations made in Section 3.2. This is a relevant outcome of this study, since following the EE principles (MEEM model) can result in many different designs that can be compared using this index.

## 4. Conclusions

This study proposed a new general and integrated conceptual framework intended to guide the design of artificial reef (AR) units by adopting an ecosystem ecology (EE) perspective, that is, by taking into account the particular characteristics of the ecosystem to be enhanced. The framework consists of three main parts. In the first one, the designer or decision maker starts from a basic initial geometry that is site-dependent and employs the Marine Ecosystem Ecology Model (MEEM) to include modifications. Logistical and structural constraints are considered during this step. As a result of following the EE principles, several AR unit designs may arise. Consequently, the second part of the framework is a new comparative and dimensionless index (*AREIT*) that allows the user to select among different alternatives by comparing their contribution to energy, nutrient cycling and shelter. Finally, once the desired design is obtained, the third part of the framework comes into play. This is a general hydrodynamic methodology to study the stability of AR units, considering the effect of extreme waves on their surfaces. As a way of validating the methodology proposed, a simple example for the particular case of Galician (in the northwest of Spain) estuaries was considered. The main conclusions from this study are:

The proposed framework appears to be a valid starting point for the design of AR units according to EE principles. Nevertheless, in real applications, additional economic, social, ecological and environmental factors could be needed.

The *AREIT* index (second main part of the proposed framework) proved to be a promising metric for predicting the positive impacts that different AR unit designs can have on a specific marine ecosystem. However, experimental and field observations, after the installation of AR units, are still necessary for several reasons. On one hand, *AREIT* provides comparative and relative results, but it does not give and idea about absolute impacts, although there is a connection between them. On the other, field observations may serve to improve the proposed *AREIT* index, in case of being necessary.

The application of the MEEM tool to the starting AR unit design for Galician estuaries led to the incorporation of an interior cavity (upper central hole) that improves the reception of sunlight. It also led to the inclusion of lateral holes that, together with the central one, favour nutrient circulation. Nest cavities were another result of applying the MEEM tool. Although these modifications provided good results for the particular case of Galician estuaries, they are likely also to be valid for other marine ecosystems. Obviously, the number and dimensions of the holes and nest cavities, as well as the initial geometry, will depend on the specific characteristics of the ecosystem to be enhanced.

The analysis of the AR’s stability under wave action is a key aspect in most areas with interest for AR operation. To this end, an accurate estimation of the wave conditions at the AR site and analysis of the wave–AR interaction by considering complex procedures are necessary. In the case of areas subject to strong currents, they should also be considered in the AR stability analysis.

The proposed AR unit design for Galician estuaries is only one of the several options that can be obtained by using the proposed framework. This solution is suitable for those ecosystems in which a balance among energy uptake, nutrient circulation and habitat enhancement is needed.

## Figures and Tables

**Figure 1 biology-11-00680-f001:**
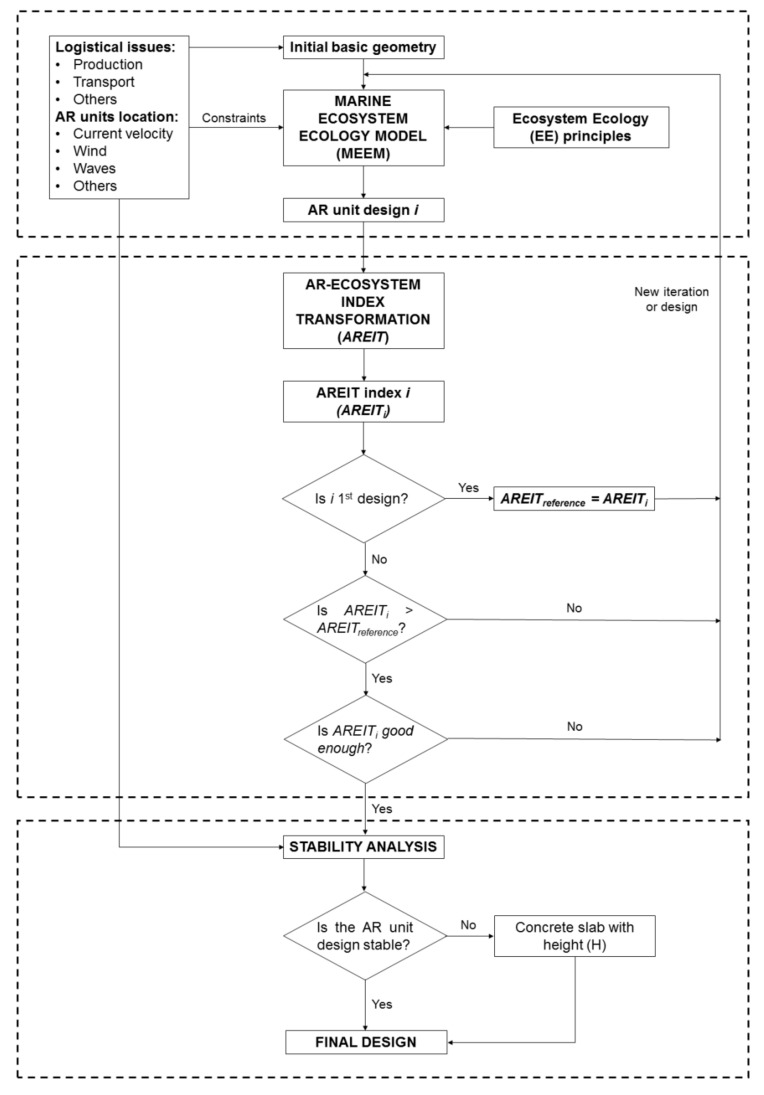
Flowchart of the conceptual framework proposed in this study. The dashed lines separate its three main parts: Marine Ecosystem Ecology Model (MEEM), AR-Ecosystem Index Transformation (AREIT), and the stability analysis.

**Figure 2 biology-11-00680-f002:**
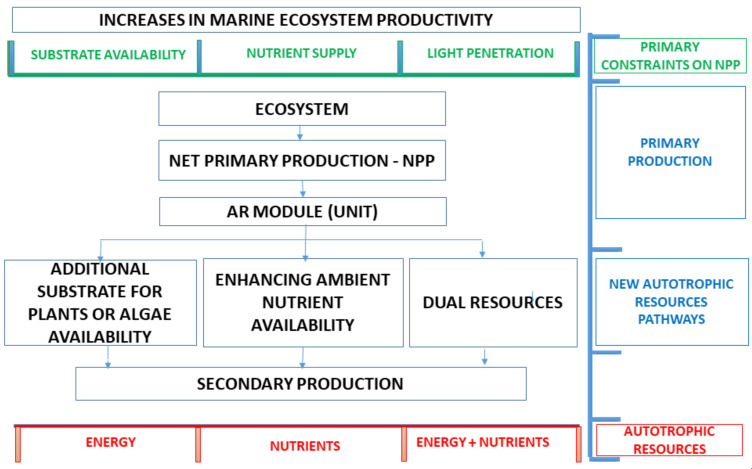
The MEEM model and its impacts on a specific ecosystem in terms of energy and nutrient availability. Black: flowchart, green: primary constraints on NPP, blue: trophic levels, red: autotrophic resource pathways.

**Figure 3 biology-11-00680-f003:**
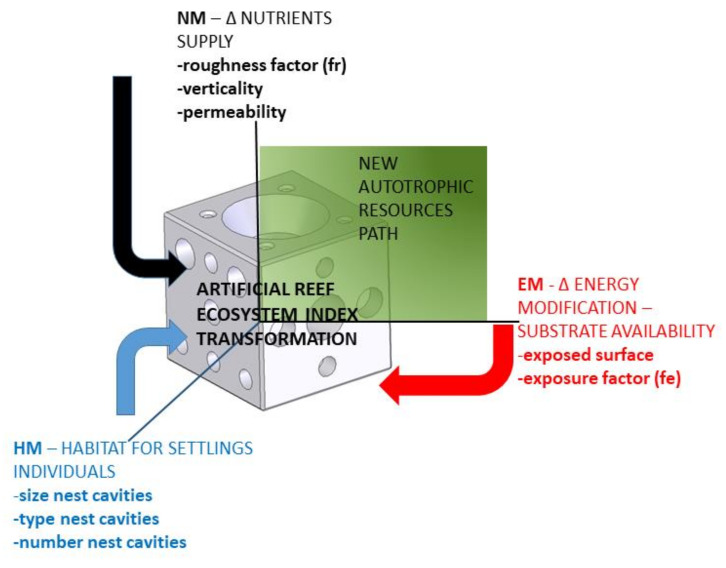
Graphical representation proposal for the AREIT partial indices: EM (energy modification), NM (nutrient modification) and HM (habitat modification) including the most important factors that affect each one of the indices.

**Figure 4 biology-11-00680-f004:**
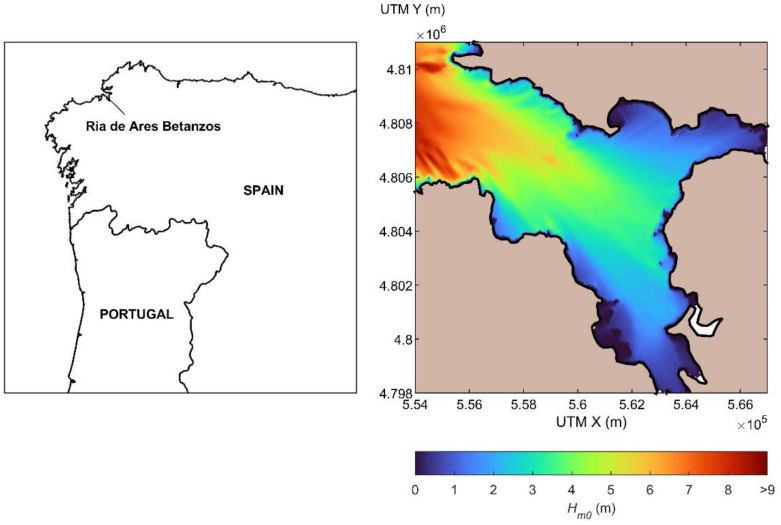
Location of the Ría de Ares-Betanzos in NW Spain (Galicia) (**left**) and spatial distribution of the significant wave height under deep-water wave conditions for T_r_ = 20 (H_s_ = 13.4 m; T_p_ = 16.1 s) and NW direction (**right**) (own source). (Simple column width figure. To be printed in colour in the online version of the paper and in grey scale in the printed one).

**Figure 5 biology-11-00680-f005:**
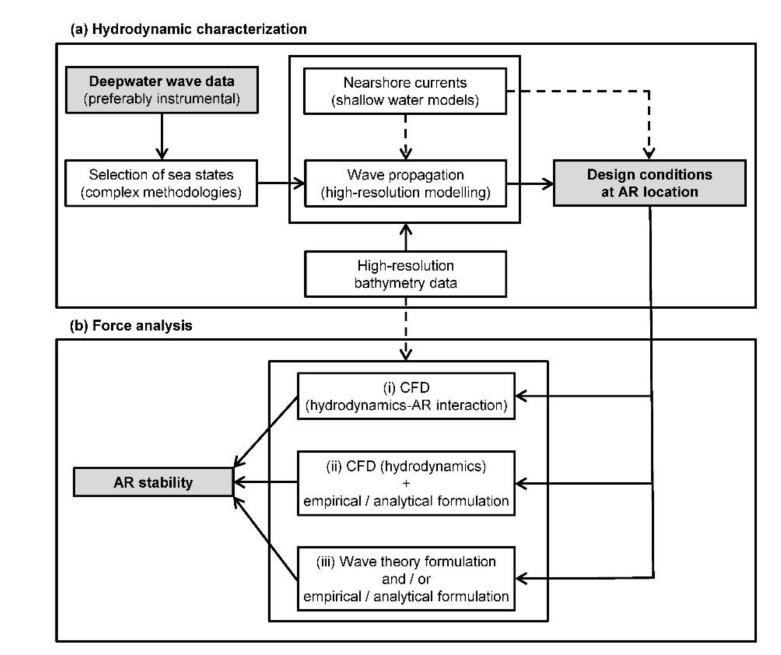
Flowchart of the process to be followed for AR stability analysis, including both hydrodynamic characterization as well as force analysis. The dashed lines indicate that the flow is only mandatory in the case of the presence of strong currents.

**Figure 6 biology-11-00680-f006:**
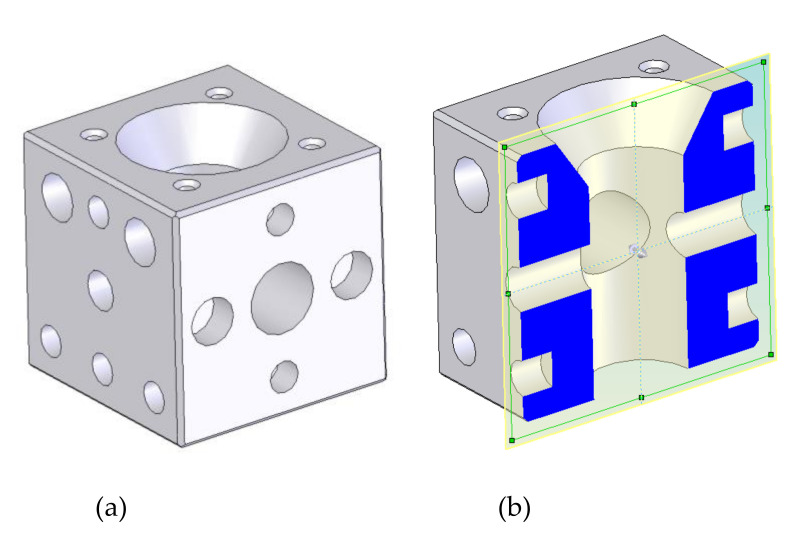
AR unit design after applying the MEEM model. It presents different types of nest cavities. (**a**) 3D; (**b**) section.

**Figure 7 biology-11-00680-f007:**
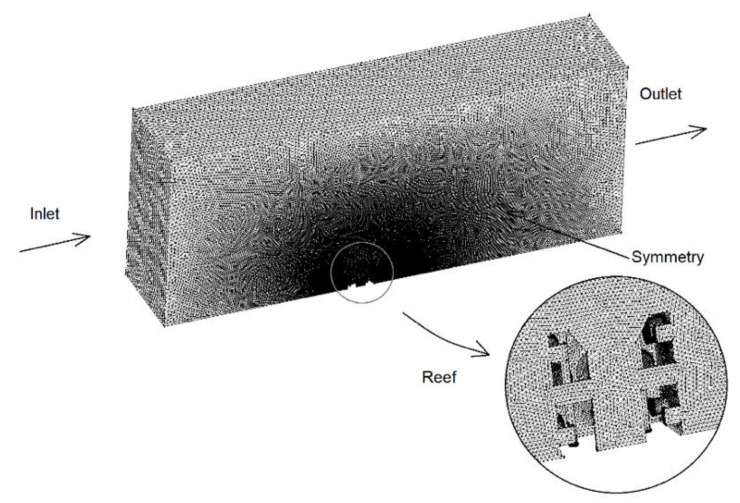
Computational mesh.

**Figure 8 biology-11-00680-f008:**
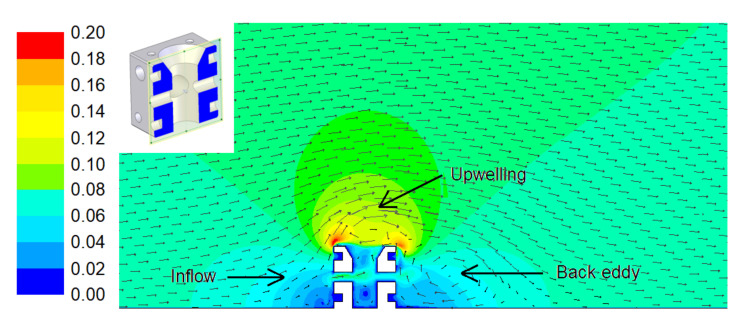
Velocity field (m/s) obtained by the CFD model.

**Figure 9 biology-11-00680-f009:**
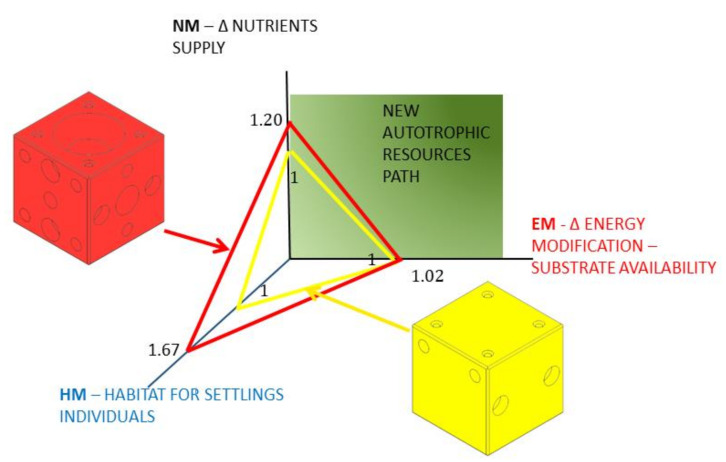
Results for the final AR design (left design, red colour) in comparison with the reference one (right design, yellow colour). This figure includes the value that each design presents for the three partial indices.

**Table 1 biology-11-00680-t001:** Correlation among the determinant factors for increasing fish biomass, the environmental parameters of the ecosystem and the pathways for obtaining new autotrophic resources. Objective: identifying the correlation among determinants for the secondary productivity of the ecosystem.

Type of Factor	General Pathways	Physical Parameters	New Autotrophic Resource Pathways
Source	Bohnsack [43]	Charbonnel et al. [44], Barnabé et al. [45], Allemand et al. [46]	Layman and Allgeier [8]
Correlation among determinants	Providing additional food	Area, material, roughness, upper central hole	Additional substrate for algae and plants availability
	Increasing feeding efficiency	Verticality, lateral vertical holes	Enhancing ambient nutrient availability
	Providing more shelter	Nest cavities (shape)	
	Providing recruitment habitat for settling individuals	Nest cavities (size and number)	

**Table 2 biology-11-00680-t002:** Main parameters of an AR unit and their potential impacts in terms of energy, nutrients and new habitat.

Determinant Factors		MEEM Model Foundations			
		Energy		Nutrients	Habitat for Settling Individuals
General Parameters	Specific Parameters	Substrate Availability	Light Supply	Nutrient Supply	
Material	AR material type	✓	-	-	-
	Material pH	✓	-	-	-
	Roughness	✓	-	-	-
Area	Area	✓	-	-	✓
Shape	Verticality	✓	X	✓	-
	Upper central hole	-	✓	-	-
	Lateral holes	X	-	✓	-
Nest cavities	Nest cavities (size)	-	-	-	✓
	Nest cavities (type)	-	-	-	✓
	Nest cavities number	-	-	-	✓
✓: Positive impact					
X: Negative impact					
-: No impact					

**Table 3 biology-11-00680-t003:** General information and relevant parameters for the final and reference AR unit designs.

**Relevant Information and Main Parameters**	Basic Design	Final Design
General information	4 vertical faces 8 nest cavities (30 cm diameter)	4 vertical inner faces with lateral holes 4 vertical outer faces with lateral holes 1 horizontal inner face with the upper central hole 12 nest cavities (20 cm diameter) 8 nest cavities (30 cm diameter)
*EM* parameters	$S_{vr}\cdot f_{ev}+S_{hr}\cdot f_{eh}=67.500\;cm^{2}$	$S_{v}\cdot f_{ev}+S_{h}\cdot f_{eh}=68.835\;cm^{2}$
*NM* parameters	$S_{upwelling\_r}=90.000\;cm^{2}$	$S_{upwelling}=107.750\;cm^{2}$
*HM* parameters	$S_{nest\_cavities\_r}=5.655\;cm^{2}$	$S_{nest\_cavities}=9.423\;cm^{2}$

## Data Availability

Not applicable.

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
