# Peer review of "Proposed Conceptual Framework to Design Artificial Reefs Based on Particular Ecosystem Ecology Traits"

_biology, 2022, doi:10.3390/biology11050680_

Round 1

Reviewer 1 Report

This is an interesting manuscript that presents an extensive analysis about the conceptual framework to assist in the design of artificial reefs according to the specific characteristics of the ecosystem to be enhanced. Moreover, the authors also include a new comparative index (AR-Ecosystem Index Transformation (AREIT)) that will assist the decision maker at the time of selecting among different AR alternatives. The document is well written scientific report and exhibits scientific solidity. Authors present the conceptual framework for the design of AR units and the AREIT index in order to serves to measure the performance of those designs that contribute to the improvement of the ecosystem in terms of energy, nutrient availability and habitat.

The title is appropriate and abstract is concise and accurately summarizes the essential information of the paper. In the methods section, the conceptual elements related to AR and environmental control variables are widely described. In addition, they present experimental and comparative results of this approach that are novel and of great interest. The conclusions drawn adequately supported by the results. Finally, this study highlights the need to consider economic, social and cultural elements in the implementation of AR

Reviewer 2 Report

  • The manuscript does not have a clearly established objective, which could be simpler than what it has been explained in text. On hand, authors mention the following sentence in the Introduction: “The main objective of this study is to present a conceptual framework to assist in the design of artificial reefs according to the specific characteristics of the ecosystem to be enhanced”. Then, in other paragraphs authors claim, “this framework will also include a new comparative index (AR-Ecosystem Index Transformation (AREIT)) that will assist the decision maker at the time of selecting among different AR alternatives”. Eventually, in other paragraph authors mention this: “a general hydrodynamic methodology to study the stability of AR units, considering the effect of high waves on the ARs’ surfaces, an aspect that was not previously included in Carral et al. [27], is integrated as the last part of the proposed framework”.
  • Consequently, I strongly suggest authors to remove the heading “Objective and main novel contributions, and simple establish the three objectives at the end paragraph of the introduction: The objectives of this work were: i) to propose a conceptual framework to design artificial reefs based on the ecosystem characteristics to be enhanced, ii) to propose a comparative index (AR-Ecosystem Index Transformation or AREIT), and iii) to develop a hydrodynamic methodology to evaluate the stability of AR units considering high waves and surfaces.
  • So, the title could change to that is established. My suggestion based on the objectives could be “Proposed conceptual framework to design artificial reefs based on particular ecosystem traits”
  • When reviewing the Material and methods section I noticed it is way too long and I recommend authors to shorten and concentrate in what it is more reliable
  • Also, I noticed that Results section is consistent but needs to reflect a more concise argument.
  • What disturbs me a lot is that authors did not include a DISCUSSION section and only a bulleted conclusions section.
  • I consider the topic covered by this manuscript is not suitable for the journal Biology but for a journal more related to ecology itself.

Reviewer 3 Report

General comments

The authors of the manuscript "Methodology for the definition of an artificial reef module: An ecosystem ecology perspective" propose a new general and integrated conceptual framework intended to guide the design of artificial reef (AR) units by adopting an Ecosystem Ecology (EE) perspective. This framework include a new comparative index (AREIT) that will assist the decision maker at the time of selecting among different AR alternatives.

Although the figures must be better explained in the captions and the topics covered are sometimes repeated, the results of this study can certainly be considered as a starting point in the design of ARs that can represent an appropriate alternatives for habitat restoration.

The manuscript is well structured showing well defined aims and scope as well as detailed "Introduction" section. The research results reported in the manuscript are clear and the English language with which the manuscript was prepared seems adequate to me. 

It follows my suggestions for author's revision:

  1. In the text the author uses the word "validated" improperly when referring to framework (line 28) and AREIT index (line 147). It would be better to use the word "applied";
  2. To improve the readiness and the soundness of the work, I suggest the authors to reduce the "Introduction" (see paragraph between the lines 78-90) and "General methodology for studying AR stability" sections, trying to avoid repetitions;
  3. The sentences between 65-71 lines can rewrite in this way : "The performance of artificial reefs was traditionally assessed through population and community ecology trough approaches based on experimental and field observations to quantify fish density, biomass, or composition, among other indicators linked to the species of interest. Nevertheless, some authors (Jayanthi et al. [13], Layman and Allgeier [6] and Liversage [14]) highlighted the need to design ARs adopting an Ecosystem Ecology (EE) perspective which allows to identify the factors that limit the net primary production (NPP) of a particular ecosystem";
  4. Since the conceptual scheme of Figure 1 consists of three parts, it is convenient to insert the topics covered in par. 3.2 within par. 3.3;
  5. Authors should provide a more detailed explanation of all the figures and tables within the captions. Moreover the Figure 3 can be deleted because it does not provide further information, Figure 2 must be cited within the text, the objective shown in table 1 must be included in the caption, Figure 5 does not report a geographical reference frame (where is Ría de Ares-Betanzos?) and the coordinate system adopted (UTM?);
  6. In paragraph 3.5 I would like to point out two aspects: (1) the limitation of large-scale models depends on the low spatial resolution which is not able to analyze the wave propagation towards the coast; (2) the downscaling procedure allows the local scale model to use the boundary conditions provided by the large scale model. transfer wave data measured at a given gauging station to a virtual station located offshore the area of interest is called geographic transposition (see Contini and De Girolamo, 1998). Please modify the text between lines 480 - 488.
  7. Line 625: please insert a reference for the OpenFOAM software
  8. On what basis was the design of the reference structure chosen? In an arbitrary way? Please specify better

Round 2

Reviewer 2 Report

I have no further comments.

Reviewer 3 Report

All the concerns has been resolved by the authors